# Bodies of Knowledge, Kinetic Melodies, Rhythms of Relating and Affect Attunement in Vital Spaces for Multi-Species Well-Being: Finding Common Ground in Intimate Human-Canine and Human-Equine Encounters

**DOI:** 10.3390/ani9110934

**Published:** 2019-11-07

**Authors:** Donna Carlyle, Pamela Graham

**Affiliations:** Department of Social Work, Education and Community Well-Being, Northumbria University, Newcastle upon Tyne NE7 7AX, UK; pamela.graham@northumbria.ac.uk

**Keywords:** affect, embodied ways of knowing, inter-corporality, interspecies intimacy, sensory ethnography, movement, rhythm, canine and equine interactions

## Abstract

**Simple Summary:**

Children’s beneficial relationships with animals are well known. Companion animals, particularly dogs, have become an integral part of family life and children’s material culture. Aside from the proven physiological benefits, there is little research about what children say about their relationships with animals and how they describe them. In this paper, we bring together both horse-human and dog-human interactions, finding common ground for understanding the complexity of human development, well-being and flourishing. Dogs in schools are fast becoming a trend in helping to support and enhance children’s learning as well as their social and emotional well-being. Studies have shown that the very presence of a dog can increase children’s concentration, executive function and behaviour. In addition, equine therapy is gaining momentum and empirical studies are showing noteworthy benefits to children and young people. However, the lack of children’s voices means that the mechanisms for these benefits are comparatively unknown and unclear. In seeking to explore this, the authors utilized a visual, sensory ethnographic approach to illuminate, illustrate, experiment and reenact how the children and adolescents related, shared spaces and multiple subjectivities with their companion horse, Henry, and classroom canine, Ted. The former, Henry, champions a programme in which young people discover what can be learned from horses about communication as they learn to ride and care for them.

**Abstract:**

In this paper, we bring together two separate studies and offer a double similitude as it were, in finding “common ground” and “common worlds” between dog–human and horse–human interactions. Appreciation of the process and mechanism of affect (and affect theory) can enable a greater understanding of child–animal interactions in how they benefit and co-constitute one another in enhancing well-being and flourishing. Studies have thus far fallen short of tapping into this significant aspect of human–animal relationships and the features of human flourishing. There has been a tendency to focus more on related biological and cognitive enhancement (lowering of blood pressure, increase in the “feel good” hormone oxytocin) such as a dog’s mere “presence” in the classroom improving tests of executive function and performance. Study A details an affective methodology to explore the finer nuances of child–dog encounters. By undertaking a sensory and walking ethnography in a North East England Primary School with Year 6 (aged 10 and 11 years) and Year 4 (aged 7 and 8 years) children (60 in total), participant observation enabled rich data to emerge. Study B involves two separate groups of young people aged between 16 and 19 years who were excluded from mainstream education and identified as “vulnerable” due to perceived behavioural, social or emotional difficulties. It used mixed methods to gather and examine data from focus groups, interviews and statistics using the Rosenberg Self-Esteem Scale. Photo elicitation was an additional source of information. This equine intervention facilitated vital spaces for social and emotional well-being. The important significance of touch to children’s and young people’s well-being suggests a need for “spaces” in classrooms, and wider society, which open up this possibility further and challenge a “hands-off” pedagogy and professional practice.

## 1. Introduction

“Life is about rhythm. We vibrate, our hearts are pumping blood. We are a rhythm machine, that’s what we are”[1].

The inimitable question posed by philosopher and dog trainer, Donna Haraway—“whom or what do I touch when I touch this dog” [2]—has been at the very core of both Studies A and B’s aims. Such an ambitious and pertinent question can be unfurled and answered through affect theory and an affective methodology. As affect can go unseen and uncognitised (Hayles [3]) exploring the mechanisms of affect and its role in child–dog interactions must be sensitive to sensory, corporeal and embodied ways of knowing. The research of Gee et al. [4,5,6,7] has provided a valuable contribution to the field of Human–Animal Interactions (HAIs), particularly child-dog encounters, and forged a strong foundation on which to build further meaningful insights of this exceptional and enduring relationship. Gee [7] calls for empirical studies which contribute to the field of anthrozoology through consideration of “small theories” and process being at the core of future directions in research that clearly demonstrate the variant benefits of dogs to children in schools and classrooms. In addition, a systematic review of the benefits of reading to dogs’ schemes have proven noteworthy in terms of enhanced literacy skills, cognitive functioning and prosocial behaviours, especially in children with attention deficit hyperactivity disorder (ADHD) [8]. McCardle et al. [9] also evoke further inquiry into this emerging field as they suggest the need to embrace qualitative, longitudinal and robust methods. As ethnographers and participant observers, we found that becoming deeply immersed in the settings alongside the children and Ted and Henry enabled a co-production of “situated knowledge” as purported by Haraway [2] and Barad [10]. This methodology encompasses sensory and walking ethnography in order to address the challenges of “capturing” affect. As affect theory has yet to reach a unified definition, this study takes the premise that affect is invisible, unseen, “felt”, and therefore elusive. It can also escape being cognitised and verbalised and remain unknown. To bring the affect of having a classroom canine to the fore, we also synthesized employing the lexicon of Ingold’s [11] “lines”, Barad’s [10] concept of “intra-action” (see Appendix A) and Deleuze and Guattari’s [12] “territories”, and “bodies without organs” (BwO) (see Appendix A), please refer to notes. By this we mean that children and young people can flourish (have positive emotions and increased self-esteem) by being enabled to explore and expand the various “territories” and spaces they inhabit. Therefore, they are enhancing their well-being through unregulated and controlled bodies (becoming bodies without organs as it were) in the creation of greater autonomy and relatedness as afforded by their close and intimate interactions with animals.

This process and synthesis emerged in surprising and illuminating ways in Study A (child-canine interactions) and Study B (adolescent-equine interactions). Indeed, the “common ground” of both studies provides not only a double similitude of HAI narratives through children’s and young people’s voices, but it also aligns with the call for community cohesion between humans and animals as purported by Taylor and Giugni’s concept of “common worlds” [13]. Fundamentally, this challenges human exceptionalism and the Anthropocene. In addition, as concern grows for children’s mental health [14] and a “rise in therapeutic education” ensues [15], it is fundamental that current research not only addresses robust and rigorous design, but also employs affective methods which can reveal new insights and understanding of what has been termed a “pet pedagogy” [16]. Before such claims can be trustworthy, a process model [17,18] is foregrounded; as such, attention to movement, touch and inter-corporality is discernible. The exploration, experimentation and enactment of child–dog and adolescent-horse interactions offer a contribution to the field of human–animal studies that elicit children’s experiences and encounters both as narrative and “visual vocabulary” [19]. In doing so, the model’s affective nature reveals important new insights around children’s perceptions of what they “touch” when they touch their classroom dog. Several authors have noted the disappearance of touch in middle childhood within school settings [20,21]. As touch and its affordances of social connectivity and well-being in child and human development are well documented [22,23], the need to consider the current classroom landscape, and wider cultural spaces, in supporting growth, learning and children and young people’s flourishing is paramount.

The idea of “flourishing” is used to expand thinking beyond happiness and highlight the need to consider it as well-being theory [24]. We concur with Seligman’s definition that “flourishing” relates to well-being of which positive emotion and positive relationships are core features. Affective and vital spaces for social and emotional well-being are animated and discussed in terms of addressing the current concern, both in schools and in wider society, for children and young people’s mental health to be improved.

## 2. Materials and Methods

The research design for Study A (see the protocol in Table 1) aligned with the theoretical framework of affect. In addition, Deleuze and Guattari’s affective (vitalism) philosophy was integrated into the protocol to elicit (visually and materially) the mechanisms of affect from child–dog encounters. By utilizing this design, through a series of workshops, the children were able to “reenact” their relationship with Ted in the classroom (visually), thereby opening up the potential to reveal and “show” affect (feelings), as they can often go unnoticed or unexpressed through language or text. This approach and this design were fundamental to answering the following research questions: (1) What do children tell us about their relationship with a classroom canine? (2) What kinds of relationships do children have with a classroom canine? (3) How does this relationship impact child well-being? In animating affect through artwork and creative media, affect was not “captured” (as its very mechanisms escape capture and representation) but is embodied, thus seen, recorded and witnessed.

### 2.1. Study A

Study A was carried out in a North East England Primary School, over two phases, with Year 4 (aged 7 and 8 years) and Year 6 (aged 10 and 11 years) children (60 in total) and their classroom canine, Ted. The children have grown up alongside Ted, a springer spaniel, and he has been in the school setting since being a puppy. He is now 3 years old. Ted joined the school following a landslide victory in a mock election that took place when the children were learning about democracy and voted to get a school dog. The Head Teacher honoured this vote and the school carefully planned Ted’s introduction into the setting, adhering to health and safety protocols and ethical consideration for Ted’s welfare and well-being. He has become an integral part of the children’s learning community, joining in school life as he wishes, such as lessons, reading time, play activities and sharing time with staff in their offices. Ted is cared for by his class teacher, and returns home with them at the end of every school day. Ted is free to move around the classroom and the children are assigned caring duties such as replenishing his water bowl and taking him for walks. He has a designated sleeping and resting area, although often chooses alternative spaces alongside the children. The children have been encouraged to learn Ted’s body language so they can ensure his well-being, for example, if he lowers his tail and seeks a separate space, they know to leave him alone and not to disturb him. They have also become attuned to his “springer sprawl” in which he will lie on his back, indicating a wish for tummy and chest rubs (See Figure 1).

The study was conducted over a full academic year. In order to apprehend the children’s experiences of their encounters and interactions with Ted, a non-representational, visual and sensory ethnography was employed. As affect often goes unnoticed and uncognitised [3], a way of understanding their experiences and relational process with a dog meant the emergence and development of a robust methodology and protocol that afforded embodied ways of knowing. This enabled the elicitation of expression through non-textual modes (creative media). The visual materialisation of the children’s lived experiences with Ted are therefore also an embodied process in itself. Thus, the limitations of language and uncovering affect (often silenced through a flat ontology) are made possible. Through describing embodied and inter-corporeal experiences, of both participants and researcher, we reveal new insights and knowledge of this significant aspect of child–dog relationships. This is a fundamental aspect of this research. It encapsulates the importance of a post-structural paradigm. Such an ontological and epistemological positioning is no less rigorous than traditional positivist approaches. Significantly, it calls for a re/”turn” to corporeal ways of knowing. Movement comes before language, infants are not pre-linguistic, and language is post-kinetic [25]. They possess a “kinetic melody” [25]. They use this fundamental and sometimes forgotten way of communication to initially draw others to themselves; in eliciting movement, they elicit care, “good enough care” as postulated by Winnicott [26]. They move their mouth, lips and tongue, along with all four limbs. In this dynamic corporality, this creates a vital energy, a soundscape, a “storm” as Stern [27] so aptly calls it. Stern’s description of the “storm” of hunger conjures up dramatically and dynamically the infant state of bodily eruption (movement and sound) in their quest for sustenance. Therefore, the importance of the body and its visceral (affective) states influence, and can be key to, our emotional well-being and flourishing.

However, in this instance, primacy is not given to language or words alone. We extrapolate an embodied and corporeal significance to the nature and essence of human–animal interactions (HAIs). We attune to movements and sounds, as Stern suggests, in synchronous and harmonious, mirrored responses from our own bodies. The inside becoming the outside, in affective states of relational attunement and emotion. These concepts help us socially navigate and map our space and place in the world. It creates a score (or lines) like a map on which the sounds and movements encountered and experienced in a territory (or space) are emergent and unfolding as patterns of interaction and interconnection [28]. We see this in the parent-child’s “dance” of reciprocity that is a choreographed display of closeness and distance [27]. It is a mutual coming together-coming apart as eloquently depicted by Erin Manning and her unfurling of the tango dance [29], referring to this rather delightfully as the dance of individuation, which is “always more than one”. In resonance with Manning through her evoking of the tango dance as an example, Deleuze and Guattari’s [12] notion of “doubling” also resonates with the corporeal dynamics of HAIs and how they afford corporeal modulation. The Deleuzian [12] example exemplifies, rather aptly, Captain Ahab “becoming-whale”—a body coupling and doubling that emerges in Ahab’s quest to find his nemesis Moby Dick. What is striking here is how this dance is also created in children’s relationships with animals, and through this paper, we endeavour to show and reveal such dynamics which are recited and enacted (visualised) by the children during encounters between them and their classroom dog (Study A) and horse companion (Study B). Like an infant’s swaddling or comfort blanket, their dog (or horse) affords opportunities of deep muscle pressure, which soothe, calm and aid self-regulation of the body and then emotions. Ted and Henry have been something akin to a “second–skin” defence, beautifully made manifest by Esther Bick [30,31], as a way of managing the uncertainties and insecurities we face outside our family system or network. Not just a transitional object (item of comfort) but a grounding, animate and dynamic human-non-human body [26,32]. The visual portrayal of the intimacy of humans and animals has captivated scholars across culture and history. Artists such as Mary Cassatt and Leonardo da Vinci have pictured in their respective works how the boundaries between human bodies and animals can and become intertwined, fused, almost merged and blurred. Cassatt depicts children with dogs as well as children and parents. Colours and contours used explicitly seem to marry child–animal and child–parent bodies’ together in the sense of a symbiotic whole. Likewise, da Vinci’s [33] intricate and detailed, often anatomical drawings of horse riders reveal a rich blend of both horse and rider bodies, where it is difficult to see where one (body) part ends and the other begins. What is evident is the embodiment of the encounter also embodied and caught on canvas with the rhythmic brush strokes belying the very image of thought. Study A’s use of “etudes” (drawing exercises) as a visual witnessing (and recording) of these intimate events also show something remarkable and similar in repeated patterns of child–dog interactions, observed in the classroom. This occurs in the same manner as ethnographer (participant observer) sketching, as embodied motion with “fingery” eyes dwelling over the “unit” of analysis, a unit of wonder and curiosity, child and dog. Visually materialising the intimate and inter-corporeal experience can be seen in both Figure 2 and Figure 3. This coupling and this doubling reflect and propel the Baradian [10] notion of “intra-action” and how child and dog mutually co-constitute one another, that is, they relate in regulatory, rhythmic and sensory ways through tactile stroking and deep muscle pressure.

This is further revealed in the “etudes” (drawing exercises) below (Figure 2 and Figure 3).

We argue, in this paper, that we need to make the body an important aspect of discourse, and by exploring bodily patterns and movements we can reconfigure childhood “attachings” in both the human and non-human world we inhabit. Cartographic-style inquiry has been utilised by the somewhat obscure and little known work of Fernand Deligny [28], a great influence on the thinking of philosophers Deleuze and Guattari. Deligny developed a novel way of working with children and young people with autism in order to engage with them “outside of language”. He was critical of the dominant psychiatric and positivist educational perspective of his day. He did not see autism as a pathological deviation from a developing “norm” and sought to practice in a way that gave attention to small gestures, arrangements of acting and living, rather than seeing personhood as foundational upon conceptions of agency and sovereign autonomy alone [28]. Such relational networks were given the Deligny neologism “arachnean”, meaning web-weaving [28]. This intriguing mode of how we consider social organisation is not only taken up by Deleuze and Guattari [12] in their concept of the “rhizome”, but it also aligns well with Sheets-Johnstone’s “animate” and “dynamic” prose to highlight the complex and subtle elements of the body as “kinetic phenomena” [25]. In bringing the body back into methodological praxis, we can begin to see in new ways and render the idea of movement (and body rhythms) as qualitatively significant to growth and flourishing. That is, the body can be given greater ontological credence as a tool or instrument of “situated” knowledge, in accordance with Haraway’s thinking [34]. Such notions of the “lines” we create through movement, walking and animation (during canter or trot in equine therapy, or walking and stroking with dogs) are further endorsed by the imminent geographer Tim Ingold [11]. It is rather apt that Ingold’s appraisal of Actor-Network Theory by Latour [35] depicted an arachnean-type ant analogy similar to the spider web of Deligny [28], as opposed to a purely Latourian mechanicist social assemblage. Indeed, “wander lines”, the term coined by Deligny’s [28] work that explored the movements of autistic children, can open up new and exciting ways in which to think about movement, touch, affect and emotion. In aligning and diffracting (breaking apart, then putting back together) theory and concepts (reading them apart, through, in and alongside one another) as purported by Smartt Gullion, we can peruse a “mapping” (diagramming) of the social milieu of child and animal (dog, or horse in this instance) [36]. Such mapping can animate and illuminate, by being in the body dynamically, a visual-material unflattening of the encounter. This affective and tactile-kinaesthetic encounter can be lost in transmission and translation (cognition) and omit significant understanding ontologically.

As shown in Study A’s protocol (Table 1), workshop activities were undertaken with creative multiple media to elicit the children’s voices about their interactions with their classroom canine, Ted. One of these workshops involved pendulum painting to enable the children’s exploration of the ideas of space and place. This process evolved and emerged into a captivating and enchanting narrative related to how the children shared the classroom space with Ted. This animation of lines seemed to encapsulate something important for the children during the process, saying how the lines were “wild” and “like a butterfly”. It culminated in the creation of a comic strip story and, through various panels, the children portrayed the relationship they share with Ted. These panels were formed by the children from a stock of photographs, which they had taken on a GoPro micro camera. Of significance was the highlighting of Ted’s “pathways” (which can be considered similar to Deligny’s wander lines). One boy stated how these “pathways bring us together and connect us” (see Figure 4).

### 2.2. Study B

A hermeneutic phenomenological approach was adopted in Study B. Although this initially seems at odds with and in contrast to a post-structuralist methodology used in Study A, what is significant and noteworthy is that we highlight our “common ground” stance that truth cannot sit comfortably within any one epistemology. An example of such is given by the post-structuralists Foucault and Heidegger and their “shift” from hermeneutics and language which could be said to be an important exploration into the very constitutive processes (of research) which produce all kinds and types of knowledge. It is on this basis that we “combine” our studies (in the sense of revelatory similar, yet different findings) in order to dynamically and epistemologically reveal new insights and understanding about HAIs. In challenging a traditional and purely positivist paradigm, we offer a double similitude, as it were, to our response to a call for studies which are methodologically qualitative and robust in design.

What we observed in our separate research studies was a striking “common ground” in not only the tactile-kinaesthetic movements and rhythms between dog–child and horse–adolescent but also the synergy and synchronicity in movement and “becoming-with” one another. This is dramatically seen in intra-actions when both human and animal bodies appear to merge, couple or double together in an intimate entanglement of closeness almost reminiscent of a symbiotic and, thus, inter/intra-corporeal encounter [12]. These entanglements seem fundamental to child well-being and flourishing, as touch in middle childhood in particular diminishes rather significantly [21]. It may even account for something similar to a nature-deficit disorder [37], tactile-kinaesthetic deficit now given sufficient credence in relation to the concept of biophilia, so poignantly described by Wilson [38] as humans’ innate need to connect with nature. Therefore, the affordances of animals such as horses and dogs, as explained in this paper, are crucial to understanding their potential to address this imbalance or deficit at key times of child development and human flourishing. More so, the safe affordances of animals could support children who have been subject to abuse (emotional, physical, sexual) and are withdrawn or tactile defensive (sensory processing and integration difficulties [39]).

In horse-human encounters, horses instinctively recognise authenticity [40]. Horses tune in to emotional and tactile “cues that may be transmitted by humans through different channels: voice, posture, expression and pheromones” [41]. Rothe et al. [42] assert that in the encounter between horses and humans, each has to recognise the other’s perspective in the relationship, which in turn “can facilitate the exploration of feelings and intuition” and enhance the understanding of “self, nature, relationships and communication.” Schultz et al. [43] highlight the particular value of relationship-building therapeutic experiences in relation to trust, communication, confidence, problem solving skills and healthier emotional relationships.

Just as Winnicott [32] used the analogy of a parent’s face serving as a mirror to help the child develop their sense of self, the horse can act as a “mirror”, reflecting human energy, behaviours (both verbal and non-verbal) and an awareness of their effects, back to the human (see Figure 5).

Fine (2015) suggests that because the horse is a large animal, in order to interact with the horse, the participant is required to find non-verbal ways to communicate in order to establish any sort of relationship [44]. In effect, the human body uses an array of multi-sensory factors, although with an emphasis on touch and physicality, to promote a feeling of confidence when confronted with a large, moving stimulus such as an equine. For example, when grooming, a human develops first-hand experience of the size difference and potential power of the horse. This mutual physical engagement also appears when leading a horse. As argued by Schultz et al. [43], the experience of being beside the horse while guiding and controlling its movements, can allow the human to move from feeling powerless, to feeling powerful. The sheer presence of a horse demands respect, and the emotional and cognitive state of the individual responds by attempting to gain their trust and respect. In therapies where the horse is ridden, this includes developing the ability to initiate the movement of an animal so large that it could potentially have the capability to severely injure if it so wanted. Ultimately, riding a horse creates a fundamental change in the positional relationship between the human and the horse. Edwards [45] suggests that riding interactions can establish and develop communication, balance and trust. These connections with horses may, as Rothe et al. [42] suggest, encourage the growth of self-esteem, which “may be increased through a new found ability to positively influence another being”. Once a relationship between a horse and a human is established, rapport begins to grow. This in turn gives a feeling of inner pride, empowerment and accomplishment [46] (see Figure 6).

The benefits of association with and positive interactions between young people and horses are multi-layered and include “care translation, socialisation and conversation, self-esteem promotion, companionship and affection stimulation” [42]. It is conceivable that these benefits can be integrated into the young person’s sense of self and transferred to other arenas of their lives. This natural ability to attune to others, to observe, evaluate and respond authentically and immediately to non-verbal attitudes and behaviours means that horses are ideal therapy and companion animals [47].

In addition, communication between horse–human is gestural and rhythmic, with dressage being perhaps the most significant example of how this process itself, similar to dog walking, is a corporeal inculcation of harmony, synchrony and rhythm. Its dynamic “rhythmic entrainment” [48] becomes a repetition of sensory affectedness and kinetic intensities calling for a body (rider or walker) to tune into its senses, transcending the body and mind dichotomy, to become what is termed rather befittingly a “rhythmanalyst”—becoming aware of one’s body rhythms and movements [49]. The elusive essence of affect can therefore become materialised in and through these studies by their interplay of both text and image. As seen in Figure 7 and Figure 8, the rhythm of riding (canter, trot) can also evoke both hypoarousal and hyperarousal states [39], thus being calming and regulatory and as well as improving attention and focus. This is also seen in Study A forming a “common ground”. This is noteworthy as these states are related to affect attunement with the dog and horse, both affording a way of relating in an inter-corporeal way.

## 3. Results

### 3.1. Affect and Rhythmicity

#### 3.1.1. Reenactments of Individuation’s “Dance” and Inter-Corporeal Encounters

By reproducing embodied ways of knowing, new knowledge has emerged. Lupton [50] discusses eloquently the primordial scene and how infants communicate pre-linguistically yet very effectively through gestures, sounds and movements. Results from both Studies A and B demonstrate embodied ways of knowing through these reenactments and, as Stern postulates, this “dance of reciprocity” is fundamental to developing a sense of “self” and individuation in the world. This perhaps also accounts for Manning’s [29] reference to individuation always “being more than one”, that in finding ourselves, it is through an “other”. This other can be non-human, as we have shown in how “mirroring” is reproduced in both canine and equine encounters. Therefore, the need for symbiosis or entanglement in our ways of relating is always a physical one, an inter-corporeal production of safety, security, reciprocity and wholeness.

#### 3.1.2. Creating Spaces and New, Shared Territory

The revival of thinking in a vitalistic and corporeal way enables the opening up and expansion of spaces for multi-species well-being. Employing visual-material methods, Studies A and B demonstrated, respectively, how qualitative inquiry can allow the essence of experience to be seen and materialized dynamically. Through the interplay of image and text, a fresh and creative element to understanding human-animal interactions is provided. By both researchers being “drawn to see” [51] and gazing long enough through their very own inter-corporeal modes of being, the proliferation of children and young people’s itineraries have been narrated and arranged. These seek to uncover alternative and significant aspects of relating that enhance well-being and human–non-human flourishing. Tactile, kinaesthetic melodies are brought to the fore [25] and reiterate the need for “skinship” [29] and touch between bodies and across species. The making of this “territory” [12] in union with one another brings bodies together as a network that can weave new pathways and patterns for relating. We have demonstrated the significance of children and adolescents not just using space but also actually producing space. In terms of relatedness, autonomy and a sense of self, this space and expansion or “bubbling” [52] of territory is relevant to what “bodies can do” [53]. It is clear that non-human relationships are important to human social and emotional well-being and flourishing. By being in synchrony with one another, “body work” [1] can enable multi-sensual flows of affect and emotion, affording mutual co-constitution and well-being.

#### 3.1.3. Developing Alternative, Post-Human, More-Than-Human Narratives

As Bessell van der Kolk aptly argues [1], “[t]he body keeps the score. If the memory of trauma is encoded in the viscera, in heart-breaking and gut-wrenching emotions, in autoimmune disorders and skeletal/muscular problems, and if mind/brain/visceral communication is the royal road to emotion regulation, this demands a radical shift in our therapeutic assumptions.” Consequently, rhythmic ideas of the relational aspects of HAIs can offer exciting new ways with which to understand and therefore enhance human and animal well-being and flourishing. In becoming “rhythmanalyst” [49,54], we are attuning to aspects of being which pay particular attention to inter-corporality. A rhythmanalysis of the data entailed showing Ted in the classroom and visually materializing his movements and patterns of HAIs (see Figure 9 and Figure 10). This non-representation analysis means that the data are not codified or categorized, thereby retaining the affective nature of experience through the use of visual-material methods. By viewing relationships through the lens of sharing a “common world” together [13] challenging the Anthropocene, we have highlighted the value of embodied ways of knowing through material-discursive practices. In taking such a stance, we have revealed alternative modes of being in the world through the conducting of affective methodologies.

Through mapping and drawing out the lines of the event [55,56] (interactions between humans-horses and humans-dogs), we can offer the image, etudes (drawings), photographs and artwork (as cartography and map, shown in Figure 11a,b as a way of stepping outside of usual assumptions and interpretive methods to depict new and exciting configurations of relationships which encompass non-human beings. These new forms of “seeing” are beyond linguistics and make visible bodily affects. This affect may otherwise be lost in transmission between bodies as merely “felt” experiences and go unknown or unspoken.

## 4. Discussion

This paper attempted to explore and uncover what is encountered and experienced outside of language and cognitive perception. In using an affective, qualitative methodology, we hope to have revealed what potentially lurks in the “unknown” and “unthought” dimensions of experience during HAIs, thus challenging Platonic and Cartesian privileging of intelligence (human). Our methods cross over and embrace differing disciplines in order to expose how opening up discussion around the politics and ethics of touch and the body can redefine our understanding of HAIs. This could offer fresh insights into the significant tactile and sensory potential to mutual well-being (of HAIs) for human and non-human flourishing. By mapping, drawing, diagramming and animating human–animal bodies and viewing them as rhythmic events and encounters, we can highlight the mutual co-constitution of benefits such as corporeal modulation (emotional regulation) and deterritorialisation–reterritorialisation (creation of space). These vital spaces, in which human–animal bodies move and intra-act in harmony and kinetic melody with one another, become vital spaces for well-being. In true Deleuzian terms, they become bodies without organs (BwO) in deterritorialised spaces, meaning they are not regulated or organized, and are static of fixed entities. They are human–animal bodies which are entangled in intimacy and empathy. This inter-species relationship is one of sociality, doubling and inter-corporeal communication on an egalitarian plane. It disposes of power (puissance), thus opening up the potential for hybrid, common world, relationships [57].

## 5. Conclusions

Both Studies A and B exemplify why children’s bodies matter. In conceptualizing the body as “unfinished”, “unbounded” and “becoming”, we concur with Horschelmann and Colls, that traditional constructions of child development need challenging. The need for studies to bring the body back into discourse is highlighted [52]. The authors stand with Darwin and his seminal works of illustrated observations of human and animal development (of emotions) [58]. Through visual-material methods, and in parallel with Darwin, we endeavour to find fresh and exciting ways with which to help children and young people alter the “inner sensory landscape” of their bodies following stress, anxiety and trauma [1]. That both Studies A and B share common ground means we have offered an alternative perspective on why HAIs are fundamental to both species. The activation of caregiving skills in the children and young people by Ted and Henry is a noteworthy outcome, which is of mutual benefit to both species (co-constitution). The potential of HAIs to therefore increase empathy in children and young people could also impact their self-esteem and well-being, for multi-species well-being. In returning to Haraway’s original question of what we touch when we touch a dog, perhaps when we touch a dog or a horse (or any other species for that matter), we touch our own “animality” in entangled empathy with them.

## Figures and Tables

**Figure 1 animals-09-00934-f001:**
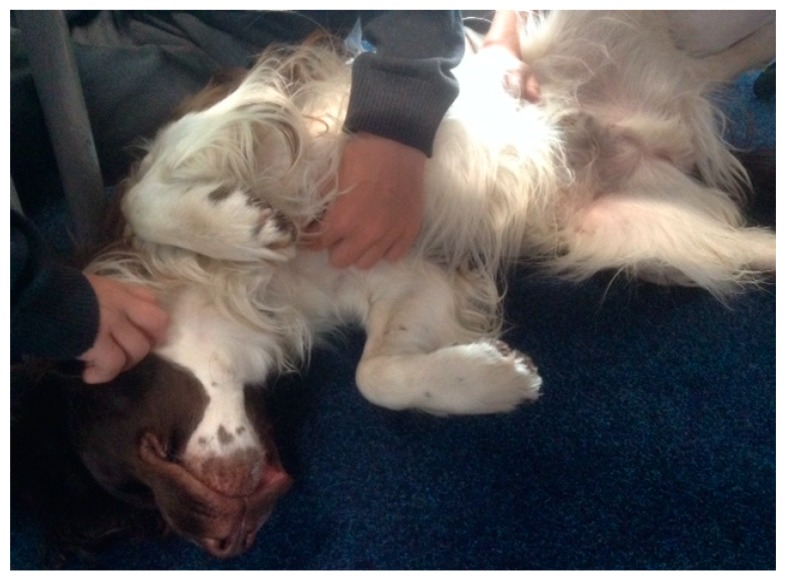
Ted’s “springer sprawl”—using his body language to “talk” and initiate tummy and chest rubs from the children.

**Figure 2 animals-09-00934-f002:**
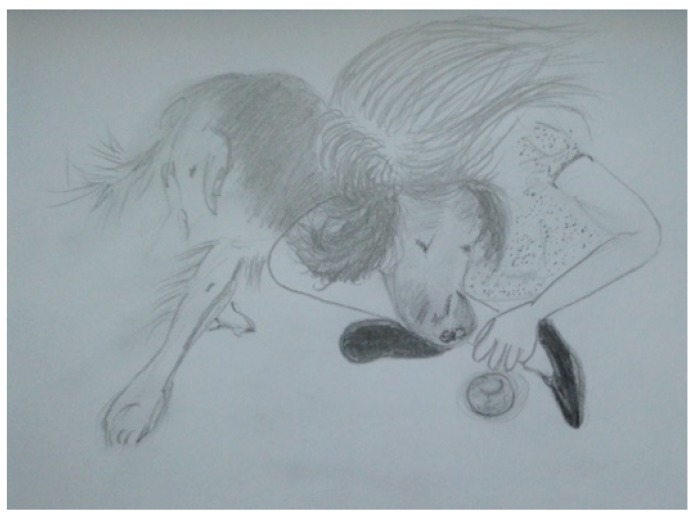
Jenny and Ted intimately intertwined in body.

**Figure 3 animals-09-00934-f003:**
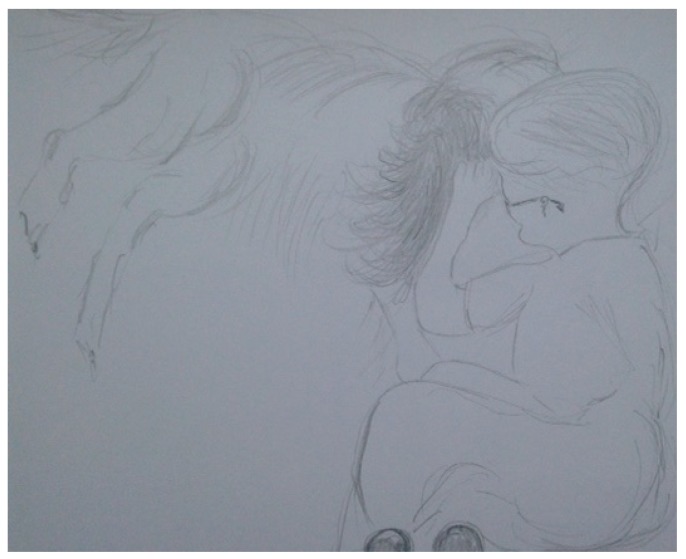
Timmy and Ted face to face with rhythmic ear stroking, and Ted responding by nuzzling into Timmy’s hair and face.

**Figure 4 animals-09-00934-f004:**
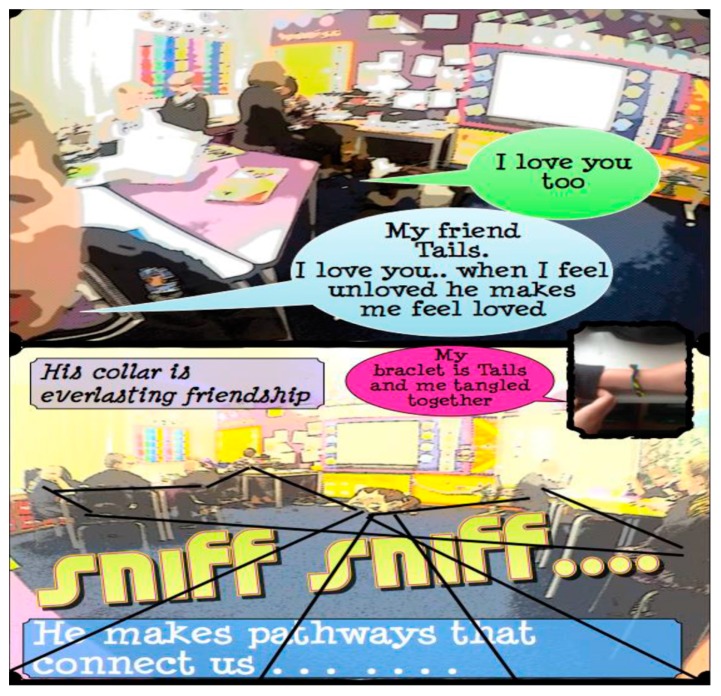
Example of comic panels showing “...he (Ted) makes pathways that bring us together”.

**Figure 5 animals-09-00934-f005:**
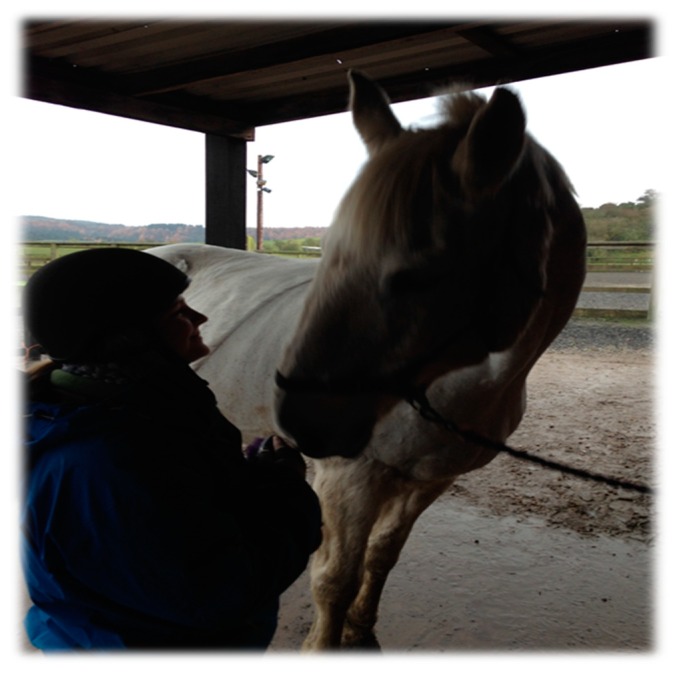
Human-horse “mirroring” and relating.

**Figure 6 animals-09-00934-f006:**
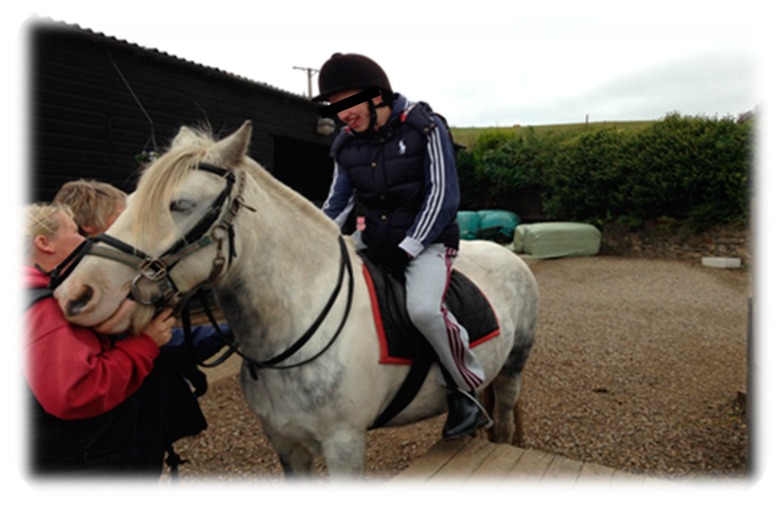
Embodied transformation. “…like a legend being on top of them. You feel so big. You feel like a boss because you’re so big. I like being on a big horse. I feel massive...I feel better.”

**Figure 7 animals-09-00934-f007:**
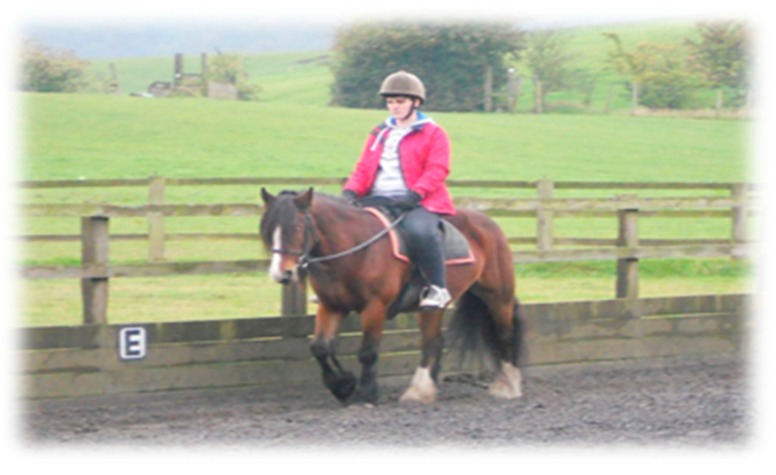
Human–horse rhythm, becoming “rhythmanalyst” (aware of one’s body rhythm) and in tune with one another.

**Figure 8 animals-09-00934-f008:**
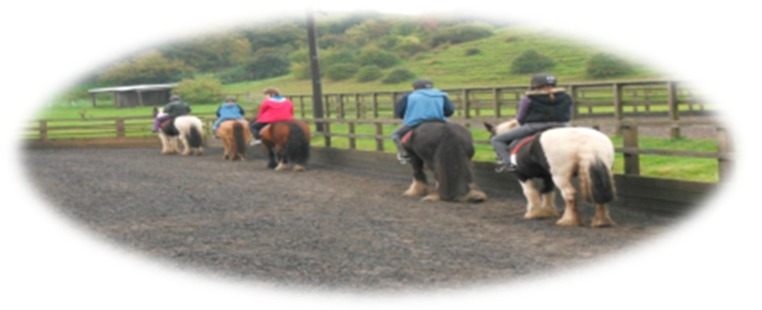
Shared and expanding spaces, creating “common worlds” and new “territory”.

**Figure 9 animals-09-00934-f009:**
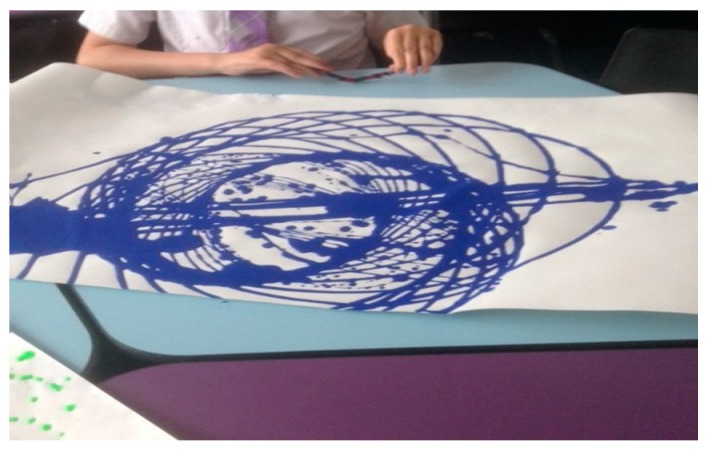
Pendulum painting with Ted. Reenacting movement in class.

**Figure 10 animals-09-00934-f010:**
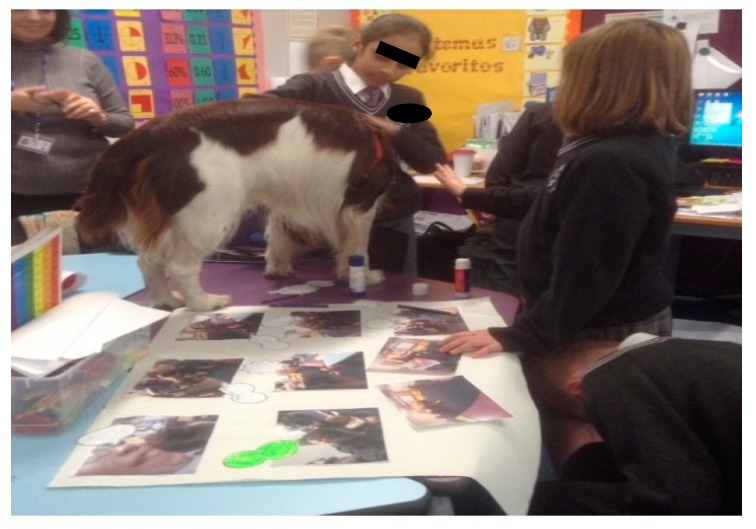
Making comic panels with Ted, telling a story with visual-material methods.

**Figure 11 animals-09-00934-f011:**
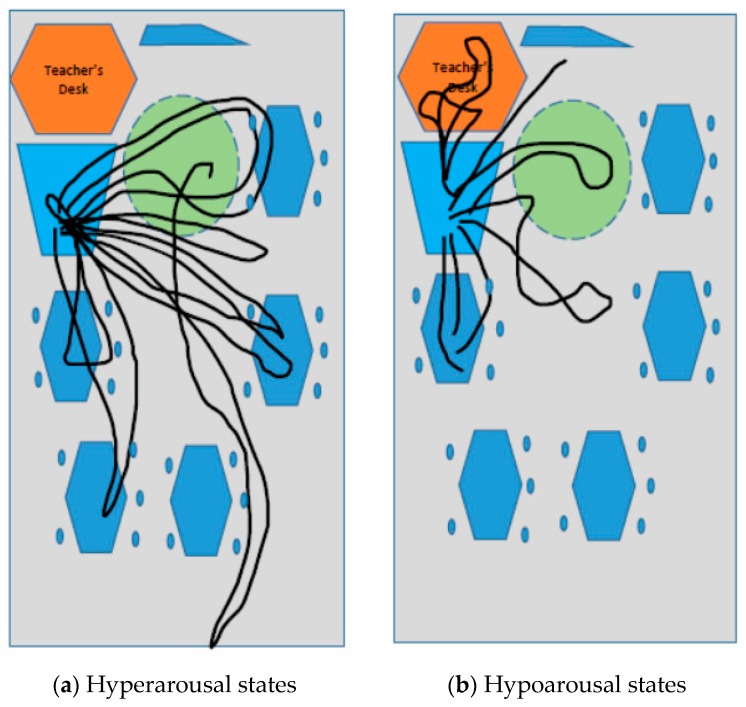
Rhythmanalysis and diagramming Ted’s “wander lines” and movement in the classroom to show affect attunement between him and the children in shared arousal states. (**a**) Hyperarousal states and (**b**) Hypoarousal states.

**Table 1 animals-09-00934-t001:** Research protocol for Study A.

What Are we Looking for?	
Movements	Embodied Experience
Activity consistent with theoretical framework	
Deleuzian actions	What we do?
1. Rhizomatic, nomadic movement (or thinking)	Mapping, diagramming. Draw lines to show mapping of walking with dog (and without) using classroom floor plan. Pendulum painting or bubble painting. Stretch, cut, colour, tear, overlay.
2. Territorialisation > deterritorialisation	Marble/ball-bearing paint and box activity. Use of string, ribbon, wool or dots to map steps and movements (child and dog) from one space to another.
3. Body without organs	Mapping child–dog movements. Drawing and diagramming shared places and space in the classroom. Use of GoPro body cam to depict lively biogeographical movement of both child and dog.
4. Smooth and striated spaces	Show texture of spaces with use of craft materials such as silk fabric, cotton wool, fuzzy felt, ribbon, foam, polystyrene, plastic (smooth) or beads, sandpaper (striated), glitter, buttons, corrugated cardboard, bubble wrap. Make swirls, waves or splashes with craft materials.
5. Lines of flight	Bursts and bolts of energy through splashing, zigzagging lines and artwork, doodling, building blocks. Moments of action like walking, dancing, building, constructing.
6. Folds	Folding of paper. Folding or bending of craft wire, pipe cleaner to sculpt body or object.

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
