# Peer review of "Bodies of Knowledge, Kinetic Melodies, Rhythms of Relating and Affect Attunement in Vital Spaces for Multi-Species Well-Being: Finding Common Ground in Intimate Human-Canine and Human-Equine Encounters"

_animals, 2019, doi:10.3390/ani9110934_

Round 1
Reviewer 1 Report
This is a fascinating paper, which will be of interest to a range of practitioners. It is clear that you have conducted two valuable studies, in two different contexts, and as such this will appeal to different audiences.
However, you must ensure that the content is accessible to such a range of readers, including non-specialists. To this end, it is very important that the paper is written to avoid overly long sentences (eg lines 30-34 are one sentence). Leaving references to the end of sentences would also assist with the flow for the reader eg lines 48/51 etc.
Also use of overly academic phrases should be minimised. For example, what do sentences on line 112-3 and 173-175 actually mean? Do you feel that this is accessible to your audience? I'm not sure that a classroom practitioner or therapist would be able to access your work in its current form. If you can address this I think that the paper would appeal, and your messages would be more widely heard.
I found that the introduction could have been structured more clearly. What were the key research questions? Affect and affect theory needed to be explained more clearly. What do you mean by 'flourish'? Cognitively? Emotionally? This needs definition.
The research design also needs explanation, for example, what are Delusian actions, and why are they focused on here?
The conclusions, and a return to research questions, plus some consideration of the implications of your study for others would also be useful. The balance of this paper needs to ensure that enough consideration is given to discussion of findings.
Proof read carefully eg apostrophes missing line 112/ punctuation 168/ spelling 285.
Reviewer 2 Report
This is a very interesting study that attempts to analyze the interaction between children and a dog or horse. It seemed as though the article could be two separate articles; one that studied the child/classroom dog interactions and one that studied child/horse interactions. Doing justice to both in one, short article was overambitious in some ways. I also found that the use of professional jargon was excessive. If the goal is to communicate with readers this could be reduced somewhat. In HAI, it is important to mention the consequences for the animal as well. If, for example, a child is stroking the dog's ears, could this not reach the point of being annoying to the animal, even if the child wants to interact this way? From an animal welfare perspective there is also the question of a dog being at school all day, every day (presumably) when dogs typically sleep more than half the day. What were the children's ages? As a U.S. reader, giving their grades in schools did not clearly convey that to me. The work certainly has merit but attempts to pack too much into a short manuscript. The level of detail on how, exactly, the data were analyzed (rather than the outcomes of that analysis) was a bit lacking, possibly as a result of space limitations. Minor point: Please check the reference note 15. Should detreminants be determinants?
Round 2
Reviewer 1 Report
The authors have made a number of changes to the article which is now more accessible and detailed. They should be commended for a swift turnaround.
Some minor text editing to still be done.